# The Role of Brand Commitment in the Retail Sector: The Relation with Open Innovation

Jose Ribamar Siqueira [1,*], Nathalie Peña-García [2], Enrique ter Horst [3], German Molina [4] and Monica Villamil [5]

1   Department of Business Administration, Pontificia Universidad Javeriana, Bogotá 110231, Colombia
2   Research Department, Colegio de Estudios Superiores de Administración CESA, Bogotá 110231, Colombia; nathalie.pena@cesa.edu.co
3   School of Management, Universidad de los Andes, Bogotá 111711, Colombia; ea.terhorst@uniandes.edu.co
4   Idalion Capital Group, London W1J 8NR, UK; german@germanmolina.com
5   Independent Researcher, Bogotá 110231, Colombia; monicavillamilp@hotmail.com
*   Correspondence: siqueiraj@javeriana.edu.co

**Abstract:** Firm–employee relationships are a prerequisite for customer–firm relationships and, consequently, to organizational success. The development of such relationships can be particularly challenging for retailers because of the complexity of the service component inherent to the environment in which they usually operate. For this reason, organizations need to align employee behaviors with the corporate brand promise so that they can perform a more active role as brand ambassadors. This issue becomes even more complex for organizations with a presence in foreign markets. This study focuses on how the adoption of in-role branding behavior by front-line employees (FLEs) can be influenced by the level of commitment FLEs display towards the corporate brand and how commitment is consequently influenced by corporate brand identity and corporate brand identity FLES' perception of their role within the organization. The object of the study was the employees of Falabella, a multinational retailer based in Chile with a strong presence in the Colombian market. Results obtained demonstrate that brand commitment positively and significantly impacts FLE brand-oriented behavior in the retail context examined. More specifically, brand identity and role clarity positively impact brand commitment, leading to a positive impact on FLE brand behavior and job satisfaction. The results of this study offer valuable insight for scholars and practitioners regarding employee brand behavior's engendering process within a retail environment in an emerging market.

**Keywords:** branding; front-line employees; retail; brand-commitment; role clarity; job satisfaction

## 1. Introduction

A brand can be described as the result of social interactions that imprint a particular value in the minds of customers and other stakeholders [1]. The direct interactions between brands and consumers often start with a search and end with the brand's disposal [2]. During this process, the front-line employee (FLE) plays the critical role of guiding customers through the final stages of their journey with the brand [3]. The direct and indirect dyadic interaction between customers and employees can help strengthen the brand image in consumers' minds at any point of contact [4]. From a strategic standpoint, employees can be used as operant resources and active brand value co-creators to help support the corporate brand and lead to a competitive advantage [5]. The delivery of a brand's promise in service organizations requires a high level of understanding of the brand on the part of FLEs and what it means for them in the execution of their roles and responsibilities [6].

The issue of corporate branding in the retail environment is complex and different from other contexts because it deals with two category-defining variables: the extensive interactions that occur between the staff and consumers, and the fact that the products being sold are usually part of a broad portfolio of different brands turning retailers into

hybrid organizations [7]. A branding strategy executed within this context must rely on a brand's capability to build relationships for the service organization. These relationships can be developed with customers and employees and can help transmit the brand's identity and values to customers and employees through different channels. This is particularly true with FLEs, who act as intermediaries between retailers and customers. The brand-oriented behavior displayed by FLEs during interactions with customers has become extremely important for organizations because of the impact it has on firm-customer relationships, customer experiences [8], and brand loyalty [9]. The behavior of FLEs attuned adequately to the brand promise can more effectively support its delivery to customers and help retailers gain a much-valued competitive advantage in the form of a superior retail experience [10].

The relationship between FLEs and customers can be analyzed from either an external or internal perspective. Externally, companies worry about customers' perception of the behavior displayed by their FLEs. From an internal perspective, firms have grasped the need to direct this behavior by determining a specific set of brand-oriented employee behaviors that can lead to the development of brand citizenship behavior (BCB) [11]. FLEs can express BCB in two ways: (1) through in-role brand-building behavior (IRBBB), strictly guided by internal brand management norms designed to align employee behavior to the corporate brand resulting in more consistent delivery of the brand promise; and (2) through extra-role brand-building behavior (ERBBB), consisting of branding behaviors that are entirely voluntary (not part of a job description) and help support the brand [12].

Employee brand-oriented behavior in the retail context should be aligned with the corporate brand, making the brand the central point for strategy development across all areas. This alignment should be a consequence of the organization's level of brand orientation [13] and its effects on branding behaviors [14]. The organizational vision of properly aligning FLE brand-oriented behavior with the retailer's corporate brand and turning FLEs into brand ambassadors has been attracting much attention lately [15,16], but how can retailers leverage internal resources to help support its development? We argue that the adoption of branding behavior by FLEs can be influenced by the level of commitment FLEs display towards the corporate brand and that this commitment is, in turn, influenced by both the corporate brand identity and FLEs' perception of their role within the organization. The ability of FLEs to deliver appropriate services and congruent brand messages is critical in the retail environment and can be affected by factors such as role conflict and ambiguity that are more prevalent in this sector than others [7]. These issues can be resolved by providing FLEs with a clear understanding of their roles, which can improve employee brand commitment and ultimately IRBBB and satisfaction with their current jobs.

With the idea of exploring the impact of a retailer's corporate brand on how its employees represent it, this study will contribute to the marketing literature in four ways. First, it explores the impact of brand identity and role clarity on FLE brand commitment. Second, it examines how brand commitment influences FLE brand-oriented behavior, operationalized as IRBBB. IRBBB was included to account for both brand and human resources characteristics that can help explain how FLE's commitment towards the brand can support it. Third, to our knowledge, the retail sector's employee branding behavior has never been explored in South American countries. This study provides a unique opportunity to explore how concepts developed in the employee branding literature can be applied to an emerging economy context. Lastly, the novel approach of applying a Bayesian framework was employed to test the proposed model. The application of Bayesian modeling presents numerous advantages over other traditional approaches because of the way it handles sample size limitations and homogeneity, potential missing data, and model specification.

This research article is structured in the following manner: Section 1 provides a literature review of the variables that comprise the study model and respective proposed hypotheses. Section 2 discusses the proposed theoretical model. Section 3 outlines the methodology used for the study and describes the data collection and analysis procedures.

Section 4 of the article discusses the results and outlines the theoretical and managerial implications of the findings, limitations, and potential avenues for further research. Lastly, Section 5 discusses the study's limitations and provides some recommendations of routes that can be explored for future research.

## 2. Theoretical Model and Hypotheses Development

### 2.1. Brand Commitment

The literature on brand commitment was primarily developed with consumers in mind. It posits that consumers who display high brand commitment can exhibit a high active interest in product or brand information, which can result from a need to confirm their brand preference [17]. When applying the concept to employees, ref. [18] (p. 266) defined brand commitment as "the extent of employees' psychological attachment to the brand, which influences their willingness to exert extra effort towards reaching the brand's goals". The brand commitment concept originated from the organizational citizenship behavior (OCB) literature, and, as a consequence, it has been used interchangeably with the term organizational commitment [19]. More specifically, the brand commitment construct is synonymous with organizational commitment and accounts for the organization-employee psychological bond [20]. Organizational commitment has been defined as "the relative strength of an individual's identification with and involvement in an organization" [21]. Various models have been developed to explain the link of employee commitment with OCBs. For example, Scholl's model represents commitment as "a stabilizing force that acts to maintain behavioral direction when expectancy/equity conditions are not met and do not function" [22] (p. 593).

In contrast, Wiener's model characterizes commitment as the entirety of employee internalized beliefs and as an antecedent for behaviors that: (a) reflects employees' personal sacrifice for the organization; (b) is independent of reinforcements or punishments, and (c) expresses personal concerns for the organization [23]. Ref. [18] suggested that brand commitment consists of three constructs similar to organizational commitment: obedience, identification, and internalization. Obedience reflects the degree of flexibility on the employee's part to mold beliefs or actions to those of a brand; identification represents employee feelings of brand belongingness, and internalization measures the degree of influence a brand has over employees' beliefs and actions. Ref. [24] (p. 381) supported the idea of a single construct measure of commitment treating brand commitment as "the degree employees identify themselves with the brand and are willing to exert additional effort to achieve the goals of the brand (affective commitment) and are interested in remaining with the service organization (continuance commitment)".

The brand internalization process and the resulting understanding of the brand depend on employees' comprehension of brand-related information. According to [25], brand understanding consists of three dimensions developed from [26] job characteristics theory. The three dimensions are: (1) employee brand knowledge perception (employee comprehension of the importance of the brand promise and its fulfillment); (2) employee perceived brand importance (employee comprehension of how brand success can help the organization achieve its goals); and (3) perceived brand role relevance (employee comprehension of their role to achieve brand success). Therefore, internalizing the brand and its promise is crucial for delivering the brand promise to the customer [27]. This can only occur when well-defined organizational brand values, practices, and behaviors are appropriately aligned with organizational efforts. Lack of clarity can disrupt the brand behavior process leading to inconsistent service delivery.

Ref. [28] examined the dimensions of commitment to the organization as antecedents of employee behavior and suggested that organizational commitment also impacts individuals' psychological attachment to organizations. Brand commitment can help instill a sense of personal competence in employees, but beyond that, it can also kindle the desire to deliver the brand promise. The degree of employee brand commitment can also serve as an indication of employee brand knowledge. Brand knowledge can help them thrive in

their roles, and when combined with a clear understanding of organizational expectations, it can result in increased employee commitment to the organization [29]. Therefore, the diffusion of brand-related information can improve employee role clarity and identification with organizational values, which can lead to improved employee performance.

### 2.2. Antecedents of Brand Commitment

### 2.2.1. Brand Identity

Brand identity is a complex construct representing a brand both visually and verbally, which communicates to customers its qualities and characteristics [30]. Brand identity is a concept better understood when examined holistically, as it is developed from the resulting interaction between brands and consumers across many different points [31]. Ref. [32] characterized brand identity as a blend of core and extended components. The core identity represents the brand's essence that remains constant regardless of market or product changes, and the extended identity deals with the personality, relationships, and symbol associations of the brand. Ref. [33] described it as a six-sided prism consisting of: physical facet, relationship, reflected customer, consumer mentalization, culture (values), and personality. Ref. [34] later segmented brand identity theory into five critical schools of thought: "corporate identity (the identity of the organization), communicated corporate identification (identification from the organization), stakeholder corporate identification (an individual, or stakeholder group's, identification with the organization), stakeholder cultural identification (an individual, or stakeholder group's, identification to a corporate culture), and envisioned identities and identifications (how an organization, or group, envisions how another corporation or group characterizes their identity or mode of identification)" (p. 879).

FLEs must correctly represent corporate brands in a service environment to be effective. FLEs represent the corporate brand when communicating with both internal and external stakeholders. The reception and processing of corporate identity cues empower FLEs to act as decoders of corporate identity signals to the customer. By doing so, FLEs can determine the strength and influence of the corporate brand identity and help build, support, and influence the brand identity by providing feedback to the organization [35]. The brand's identity is fundamental within this context, as it enables FLEs to develop a better understanding of the firm. This can also occur through its visual identity represented by symbols and logos [36], allowing this understanding to shape their behavior when interacting with customers. Ref. [37] argued that beyond transmitting information to its employees, an organization's visual identity system also indirectly sends out information about itself to the exterior world through its employees. The literature on corporate branding acknowledges how FLEs can influence customers' brand perceptions through service development and delivery [5], posits corporate branding as means to achieve differentiation [38] that can contribute to the development of a positive corporate reputation [39], and as a tool that can help align the organization around a core brand identity [40]. Brand alignment is particularly challenging because brand knowledge tends to be asymmetrical among brand stakeholders (marketers, consumers, and channel members) [41]. When examining FLEs from the perspective of an organization's corporate identity audience, a significant portion of the organizational knowledge results from physical and behavioral cues that spontaneously occur within the organization [42] and formal internal branding programs.

Employees can also transmit psychological signals that represent attitudes and behaviors characteristic of their organization during a service encounter [43]. The intensity of these signals can be as strong or weak as the degree to which the customer identifies with the core characteristics of the organization [44]. Employees committed to a company display a higher self-appreciation and appreciation of their work, and as a consequence, the satisfaction and motivation to perform their work increases [45]. Due to the need for employees to emotionally internalize brand values in order to deliver the brand promise [46], it is hypothesized that:

**Hypothesis 1 (H1).** *Brand identity positively impacts FLE brand commitment.*

### 2.2.2. Role Clarity

Role clarity is defined as "the level of clarity an employee has of their role as a result of having brand knowledge" [47] (p. 946). Role clarity represents the employee's understanding of the brand's expectations and their responsibilities. In order to behave according to a company's brand standards, employees must clearly understand their roles, as lack of clarity can result in wasted efforts and underperformance. Role clarity can be supported by a supervisor's feedback and can ultimately influence employee job performance. The main objective of role clarity is to stimulate a type of goal-oriented behavior in employees that can be aligned with the company's brand expectations [48]. A transparent communication process designed to apprise employees about service offerings, needs and wants of clients, product features, service benefits, and corporate goals can further understand their roles within the organization [49]. Ref. [50] argued that this type of information is required to properly align employees' attitudes and behaviors with organizational goals because of how it can impact individual behavior. Additionally, there is a need on the part of employees for psychological safety within the work environment connected to fear of negative consequences to self-image, status, or career [51]. Psychological safety can be affected by organizational processes and norms, and the existence of a supportive and trusting management team can help employees feel psychologically safe and satisfied with their jobs [51]. The relationship between employees' understandings of role requirements and their satisfaction is supported in the role clarity literature [52].

The successful implementation of internal brand management practices requires the organization to acknowledge the fact that "employment represents an exchange process whereby the provision of material and socio-emotional benefits by the organization is exchanged for employee effort and loyalty" [47] (p. 942). It is also essential for management to better understand employees' needs and wants regarding their roles and responsibilities because it offers management the opportunity to better align the dissemination of knowledge according to what they learn. Ref. [53] showed that by encouraging employees to agree on an appropriate style of brand supporting behavior, an organizational leader could play a crucial role in developing a shared service brand. This can help eliminate role ambiguity and build stronger working relationships between group members through knowledge dissemination. In turn, knowledge dissemination can help employees better understand the overall brand strategy and management's rationale regarding employees, customers, and service [54]. Consequently, it can help reduce employee role conflict [55] and increase role clarity [56]. Thus, it is hypothesized that:

**Hypothesis 2 (H2).** *Role clarity positively impacts FLE brand commitment.*

### 2.3. Consequences of Brand Commitment

### 2.3.1. In-Role Brand Building Behavior (IRBBB)

Ref. [57] was among the first to point out that employee performance affects customer satisfaction and retention, and brand image. The existing research on employee branding is mainly directed at behavioral effects or its management. The research line examining behavioral effects of employee branding focuses on its relationship with customers and brand image. It seeks to understand how customers perceive FLE in terms of brand image and brand-building performance. Conversely, the line that focuses on employee branding management is procedural and focuses on requirements necessary to implement a predetermined brand personality into the building behavior exhibited by employees [58], which is the theme examined in this study.

Ref. [59] (p. 68) define employee branding as "the process by which employees internalize the desired brand image and are motivated to project the image to customers and other organizational constituents". It was not until recently that the study of the benefits of internal marketing, strategy formulation, and management started to attract attention [6,60].

There is ample agreement that internal marketing is an instrumental component that can be used to help guide employee behavior through a better understanding of brands. Ref. [12] proposed a model that specifically examined the development of employee branding behavior in the telecommunications sector. The model is based on the two forms of service behaviors that employees can exhibit: in-role and extra-role behaviors. IRBBB is defined as "front-line employees' meeting the standards prescribed by their organizational roles as brand representatives (either written in behavioral codices, manuals, display rules, and so forth or unwritten)" [12] (p. 123). Conversely, extra-role behaviors are optional and should not be included in a formal job description [61].

IRBBB can be engendered, supported, and managed internally by the firm and should be cultivated among employees to improve its relationship with customers and differentiate itself from the competition. Nevertheless, the management of IRBBB presents its own set of challenges. One of the biggest challenges has to do with the nature of the construct and its reliance on processes clearly defined internally by organizations that are, for the most part, performance-oriented. This subjects FLEs to performance reviews based on clear sets of role expectations grounded on a reward and punishment system that can impact FLEs' job satisfaction. For this reason, some authors have highlighted the need to coordinate the efforts of marketers and HR managers to support branding programs [62,63]. This combined internal organizational effort can help organizations improve the extent to which employees buy into the proposed values and norms and fully live the brand [20]. The act of living the brand is vital because by belonging to an organization FLEs gain specific understandings that help them navigate the organizational structure [64]. This act also indicates the existence of a psychological relationship between the FLE and the firm expressed by informed decisions made by the FLE that ultimately can influence organizational representation and brand commitment of FLEs [65]. Due to the role brand commitment plays in the will of FLEs to represent the brand, we propose that:

**Hypothesis 3 (H3).** *Brand commitment has a positive impact on IRBBB.*

### 2.3.2. Job Satisfaction

In order to be successful at employee branding, organizations must develop a clear understanding of the employer-employee relationship [59], which starts with the development of a psychological contract. A psychological contract is established when a new employee joins an organization [66]. This contract is based on expectations established between the two parties and is highly dependent on messages employees receive about the organization from the moment they are recruited through their tenure with the company. If there is a breach of contract, employees might start displaying diminished loyalty, negative word-of-mouth, decreased productivity, and ultimately termination of employment with the organization [67]. The psychological contract is one of the foundations of the successful implementation of employee brand image [59]. Employees have also been considered internal customers, with employee satisfaction perceived as vital to satisfying customers [68]. Job satisfaction is an expression of an individual's total evaluation of a firm according to his/her personal experiences over time [69]. Some authors argue that besides the obvious benefits gained by companies that prioritize employee job satisfaction (organizational commitment and loyalty, lower turnover), job satisfaction can also impact customer satisfaction [70]. Job satisfaction has been shown to affect customer engagement and the exchanges between customers and FLEs [70], causing companies to prioritize and invest significant resources in FLE job satisfaction. The positive relationship between these two constructs has been well-documented in various studies [71,72].

Ref. [73] suggested that one of the benefits of job satisfaction is fostering individual cooperation and contribution sentiments in an employee when performing in a team environment. Nevertheless, a direct positive relationship between job satisfaction and traditional job performance measures has still not been unanimously supported by empirical research [74]. However, it has been argued that there is a positive relationship

between satisfaction and other more formal performance measures in the form of in- and extra-role behaviors [75]. Through social exchange theory [76] suggested that this might be due to employees' desire to reciprocate when satisfied with their jobs. This reciprocation can take the form of intention to stay with the organization and other behaviors such as organizational citizenship. The more traditional job performance measures present in job descriptions and standard operating procedures increase the likelihood of employee reciprocity occurring as citizenship behaviors [77]. Job satisfaction is the most frequently examined correlate in organizational citizenship behavior studies [78]. Due to the positive relationship documented between brand citizenship behaviors and job satisfaction in the marketing literature, we posit that:

**Hypothesis 4 (H4).** *Brand commitment has a positive impact on FLE job satisfaction.*

*2.4. Conceptual Framework*

Employee brand engagement is considered as one of the main levers of organizational brand equity [47]. Nonetheless, not much research examining the effect of cognitive, affective, and behavioral aspects of internal branding in the enhancement of employee performance is available. Front line employees have become so crucial to the corporate branding effort that some authors acknowledge them as the service brand itself because of their role in turning corporate identities into reputations [79]. The role of brand identity is therefore critical in the IRBBB engendering process. According to Social Exchange Theory [80], when an organization takes up an active role to satisfy its employees' brand knowledge needs and adequately explain their roles as brand ambassadors, FLEs are able to perform their in-role brand duties more effectively. Therefore, the development of employee brand-oriented behavior should be aligned with the corporate brand, and the brand should be treated as the central point for strategy development across all areas. This should be determined according to the level of brand orientation both desired and displayed by the organization [13] and how it currently affects the implementation of branding behaviors [14]. Within this context, corporate brand management determines how the brand should be positioned to support the delivery of the corporate brand's promise and the resulting consistency required for communications and decisions across the organization [81].

Ref. [82] argue that the brand-oriented approach can guide and manage other critical brand strategy components, such as brand commitment. The delivery of the brand promise relies on FLEs commitment to the brand to support product/services performance and the delivery of the brand promise [83], and according to [84], total brand commitment is required from all employees in order for an organization to differentiate itself based on its corporate brand. Organizations that manage brands as strategic assets tend to display high levels of brand orientation, resulting in the development of strategy, culture, and organizational performance aligned with the corporate brand [85]. Therefore, internal branding is required for an organization to properly align employee behavior with its brand identity [86].

Ref. [87] suggested that brand commitment should be treated as a key attitudinal outcome of internal branding. This was based on the idea that cognition can lead to emotional responses and subsequently lead to coping activities [88], introducing an interesting avenue for the exploration of the cognitive antecedents of affective behaviors, such as FLE brand commitment [88]. Given the role that brand commitment plays in the engendering of employee branding behavior, this research investigates the nature of the relationship of some of its antecedents at the FLE level and how it is supported within the retail environment. The proposed relationship between brand commitment and IRBBB is supported by recent findings presented by [60], who argued that a deep-level bond to a firm's brand might lead to higher levels of brand identification among its employees and greater engagement in brand behavior. The development and management of employee IRBBB pose challenges for retailers. The search for a model examining the impact internal

variables can exert on the development of IRBBB has become an important topic lately for companies seeking to provide customers with a more manageable and consistent customer experience aligned to the brand promise [89–92]. Such a model is of particular importance to organizations because besides the need to know if their employees are behaving according to the organization's branding norms, they need to understand and quantify what is currently driving such behavior. The proposed model shows that FLE brand commitment within a retail environment is influenced by the firm's brand identity, which should be the starting point for any branding effort. It goes beyond the traditional boundaries of the branding literature and, from a human resources perspective, also examines the impact role clarity has on the development of brand commitment. This is important because a thorough understanding of the brand's identity and clarity regarding their roles in delivering the brand's promise can significantly impact brand commitment. Role clarity was argued to be positively related to in-role behavior [93] and more recently examined by [94], who found that it may help explain the effect of internal branding on in-role brand behavior. A firmly committed FLE who fully understands the brand's identity and his/her role in delivering the brand promise will be more willing to perform the norms of behavior prescribed by the brand and be more satisfied with the job. Figure 1 summarizes the conceptual framework proposed by the authors, and Table 1 describes each proposed relationship as discussed in the literature section in terms of antecedent behaviors (Sections 2.2.1 and 2.2.2) and consequences of brand commitment (Sections 2.3.1 and 2.3.2).

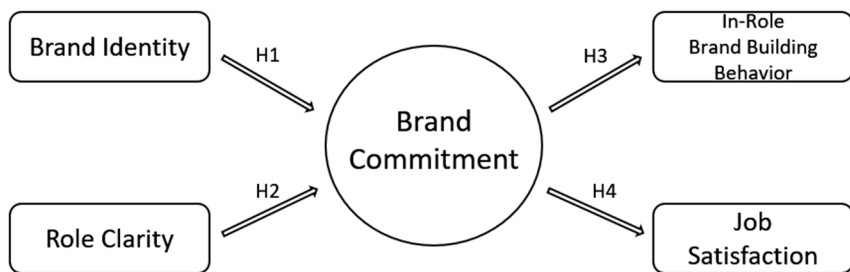

**Figure 1.** Antecedents and consequences of brand commitment of front-line employees proposed theoretical model.

**Table 1.** Proposed hypothesis.

| Hypothesis | Description |
| --- | --- |
| H1 | Brand identity positively impacts FLE brand commitment |
| H2 | Role clarity positively impacts FLE brand commitment |
| H3 | Brand commitment has a positive impact on in-role brand building behavior |
| H4 | Brand commitment has a positive impact on FLE job satisfaction. |

## 3. Methodology

### 3.1. Research Context

3.1.1. The Role of FLEs in the Retail Sector

One of the main reasons behind the increased importance attributed to brand experience in the literature lately has to do with the existing inseparability of service-oriented deliveries and the physical retail setting [95]. Branding can play an active role in shaping customers' perceptions of a product or service, but beyond that, it can also shape equivalent perceptions in employees [57]. This can happen because a brand is a representation of the relationship between an organization and its customers and employees [96]. Therefore, the need to align FLEs' behavior with the brand values has become more evident lately [97]. This can be attributed to existing discrepancies identified in the performance of employee brand-oriented behavior in service contexts that can prevent the successful management of performances and brand experiences to which customers are subjected [98]. This is easily observable within the retail environment, where FLEs facilitate the interaction between the

brand and the customer through its service. In this particular context, FLEs are directly responsible for delivering services and goods to customers [99].

FLEs have displayed a tendency to adopt one of three existing service encounter models [100] in a retail context: (1) to provide customers with what they request in an efficient and courteous manner; (2) to strive for immediate objectives accomplishment (such as sales goals); and (3) to establish a relationship with customers that can be mutually beneficial. The co-creation of the experience during the sales encounter can significantly impact service quality and ensure customer satisfaction and loyalty [101]. Service encounters can go beyond shaping a customer's perception of the service delivered [102] and help shape the customer's quality level [103]. In this particular context, the manner in which FLEs behave can significantly impact how a customer perceives the quality of a service [104], its value, and the resulting customer satisfaction [105]. FLEs transmit psychological signals during the service encounter, which are expressions of attitudes and behaviors attributed to their corporate brand [43]. The force of these signals is proportional to the degree to which a customer identifies with the core characteristics of an organization [44]. Therefore, it can be argued that service encounters with customers in a retail context are critical for the performance of an FLE because of the direct impact they can have on the organization's profitability.

Even though the role performed by FLEs in a service environment can also have a significant impact on the experience provided to a customer [99,106], it is still difficult for service organizations to accurately predict all appropriate behaviors that an employee should display to support organizational success. This might occur because many of these behaviors cannot be entirely controlled by the organization [12,107]. Consequently, the development of a method designed to fully engage employees in a brand-building process has been a challenging proposition. Many references to terms that fit the description of employee branding behavior exist in the literature (i.e., brand ambassadors, brand champions, etc.), but a widely agreed conceptualization past ensuring high-quality service delivery has not been developed yet [12].

### 3.1.2. Retail in Emerging Markets

An unsaturated market's appeal remains one of the main reasons behind international expansion [108]. For this reason, studies conducted in emerging markets can help researchers better understand how existing marketing practices and perspectives prevalent in these markets differ from those implemented in developed economies. Some widely accepted concepts developed in developed economies, such as branding, customer centricity, brand equity, and product positioning, for example, cannot be fully implemented in the existing retail context of emerging economies [109]. This might be due to existing differences between emerging and developed markets that can hinder the prescribed application of marketing theory, strategy, policy, and practice designed with developed economies in mind. According to [109], these differences can be grouped into the following five market dimensions: existing heterogeneity in the market, sociopolitical governance, unbranded competition represented by local market players, a persistent shortage of resources, and the presence of inadequate infrastructure to support businesses [109].

Latin American markets present unique sets of intricacies. In some cases, apparent consumer behavior irrationality can challenge the growth of the formal retailing sector in specific markets. In other cases, the majority of sales of the retail sector can originate from small-scale retailers. Some of these behaviors can be explained by certain advantages that make smaller retailers attractive in local consumers' eyes, such as the perception of a more personalized retail experience [10].

### 3.1.3. Study Unit: Falabella in Colombia

Falabella is a Chilean retail holding company founded in 1889 that operates department stores, home improvement home centers, supermarkets, and hypermarkets, employing over 105,500 people across markets. It has a strong presence in Colombia, where it has

been operating since 2006. Colombia is an emerging market that shares many similarities in retail structure and consumer behavior with other Latin American emerging markets. Consequently, many of the country's main cities started to witness the arrival of modern retailers such as supermarkets, hypermarkets, and department stores, among other formats. Colombia's retail reality is still quite different from the one observed in more developed economies such as Europe and the United States, where between 10 to 20 percent of the mass consumption market occurs through small retailers.

Falabella is also active in the financial, insurance, and real estate sectors. All of its different operations are present in Chile, Argentina, Peru, and Colombia. Its first international store was opened in Mendoza, Argentina, in 1993. In 1995, it entered the Peruvian market by acquiring the local chain Saga. Falabella currently owns and operates 33 stores in Chile, 7 in Argentina (Mendoza, Rosario, Córdoba, San Juan, Buenos Aires, and 15 in Peru (Lima, Arequipa, Trujillo, Chiclayo, Piura, Cajamarca, and Ica). It opened its first store in Colombia in November 2006 at Centro Comercial Santa Fe de Bogota. As of 2017, Falabella operates 25 stores in Colombia (12 in Bogota, 1 in Barranquilla, 4 in Cali, 2 in Medellin, 1 in Cartagena, 1 in Pereira, 1 in Villavicencio, 1 in Ibague, and 2 in Bucaramanga) [110].

Falabella's strategic focus revolves around female consumers. Its vision is to be women's favorite retail brand [110]. For that reason, most marketing campaigns are developed with women in mind, even though all stores also have sections dedicated to men, shoes, beauty, children, appliances, and home decoration. In order to support its positioning, Falabella employs a differentiation strategy based on safety and quality. Human resources (HR) play a very active role in the company through employee training. It works closely with its internal communications department to ensure all employees are always aware of new brand initiatives and promotions. According to the ranking published by Great Place To Work [111], Falabella was ranked as the 21st best company for employees in South America.

### 3.2. Sample

Data collection was conducted among Falabella's associates in Bogota, Colombia. The chain employs over 7000 people in Colombia. For logistical reasons, the survey was conducted only in Bogota stores. In order to avoid operations disruption, the HR department of the company opted to request employees' voluntary participation either before or after their work shifts.

Respondents were asked to anonymously express their agreement or disagreement with several statements on a five-point Likert scale ranging from 1 = strongly disagree to 5 = strongly agree. The final sample description is presented in Table 2. Answers were later evaluated according to completion and engagement. Responses submitted with incomplete answers and with a standard deviation lower than 0.3 were eliminated. The survey produced 400 completed forms and resulted in 392 usable responses.

### 3.3. Research Instrument

The survey instrument was developed as a composite of existing scales that demonstrated reliability and validity. It was then translated into Spanish by native speakers and back into English by native speakers within the framework of collaborative and iterative translation [112]. The final survey instrument reflects a comprehensive literature review, and academic colleagues also assessed its items for content and face validity. Market characteristics were taken into consideration by testing the instrument with 20 consumers to ensure the response format and that the clarity of the instructions fit the local market context. All items were rated on 5-point Likert scales and are presented in Table 3 along with references.

**Table 2.** Sample description.

| Gender | | Type of Employment | |
|---|---|---|---|
| Male | 40% | Full Time | 83% |
| Female | 60% | Part-Time | 15% |
| | | Weekend | 2% |
| Age | | | |
| 16–24 | 13% | Position | |
| 25–31 | 51% | Supervisor | 45% |
| 32–47 | 34% | Employee | 55% |
| 48 and over | 2% | | |
| | | Seniority | |
| Average Daily Interactions with Clients | | Less than 1 yr | 24% |
| 1–5 times | 25% | 1–4 yrs | 32% |
| 6–10 times | 21% | 5–7 yrs | 27% |
| 10–20 times | 17% | 8–14 yrs | 17% |
| All-day long | 37% | | |

**Table 3.** Measurement Items.

| Latent Variables | Measurement Items | Item | References |
|---|---|---|---|
| Brand Identity | I know the core components of the (company name withheld) brand. | BID1 | [35,113] |
| | The description of our mission statement is understandable. | BID2 | |
| | The description of our mission statement is easy to memorize. | BID3 | |
| | The description of our mission statement is convincing. | BID4 | |
| | There is total agreement of our mission across all levels and business areas. | BID5 | |
| | Our company transmits a consistent visual presentation through facilities, equipment, personnel, and communications material. | BID6 | |
| | Our consumables (e.g., e-mails, letters) are designed to match the overall visual elements/image of our company. | BID7 | |
| | The company's values and mission are regularly communicated to employees. | BID8 | |
| Role Clarity | I knew what was expected of me on my team. | RCLTY1 | [114–117] |
| | I felt that I had sufficient time to perform. | RCLTY2 | |
| | I know what my responsibilities are. | RCLTY3 | |
| | I feel certain about how much authority I have. | RCLTY4 | |
| | Clear, planned goals and objectives exist for my job. | RCLTY5 | |
| | My work objectives are always well defined. | RCLTY6 | |
| | I know exactly what is expected of me on my job. | RCLTY7 | |
| | Explanations are clear about what has to be done. | RCLTY8 | |
| Brand Commitment | I will work harder than I am expected in order to make Falabella successful. | BCOMMIT1 | [18,118] |
| | I am proud to work for Falabella. | BCOMMIT2 | |
| | I feel very loyal to Falabella. | BCOMMIT3 | |
| | I talk about Falabella to my friends as a great company to work for. | BCOMMIT4 | |
| | I really care about the future of Falabella. | BCOMMIT5 | |
| | My values are similar to those of Falabella. | BCOMMIT6 | |
| | I feel like I really fit in at Falabella. | BCOMMIT7 | |
| In-Role Brand Behavior | In customer-contact situations, I pay attention that my personal appearance is in line with our corporate brand's appearance. | IRBBB1 | [12] |
| | I see that my actions in customer contact are not at odds with our standards for brand-adequate behavior. | IRBBB2 | |
| | I adhere to our standards for brand-congruent behavior. | IRBBB3 | |

**Table 3.** *Cont.*

| Latent Variables | Measurement Items | Item | References |
|---|---|---|---|
| Job Satisfaction | Please select how satisfied you are about job security working for Falabella. | JSAT1 | [119,120] |
| | Please select how satisfied you are about physical conditions (e.g., safety, break rooms, etc.) of this company. | JSAT2 | |
| | Please select how satisfied you are about fringe benefits working for Falabella. | JSAT3 | |
| | Please select how satisfied you are about the pay you receive for your job working for Falabella. | JSAT4 | |
| | Please select how satisfied you are about the recognition that you get when you do a good job working for Falabella. | JSAT5 | |
| | Please select how satisfied you are about the freedom you have to do the best you can at job working for Falabella. | JSAT6 | |
| | Please select how satisfied you are about opportunities for career advancement working for Falabella. | JSAT7 | |
| | Please select how satisfied you are about the type of work you do for Falabella. | JSAT8 | |

### 3.4. Method of Analysis and Measurement Model Analysis

Each construct can be seen as a latent random variable built from the information available from the items presented in Table 3. Consequently, we propose a Bayesian hierarchical multinomial ordered probit latent variable approach to address the associations between the latent constructs [121]. This approach, described in more detail in the Appendix A, has been explored successfully in similar scenarios [122,123].

The hypothesized associations (Figure 1) between the constructs are assessed as part of the model estimation, which is performed using Markov Chain Monte Carlo (MCMC) methods [124]. The Bayesian approach does not rely on the central limit theorem for estimation, and it is, therefore, robust for low sample sizes [125]. It also aligns with the American Statistical Association's recent recommendations regarding summaries of evidence, moving away from p-hacking risks and traditional *p*-value-based interpretations that may lead to inappropriate assessments of evidence [126,127].

Each item constitutes a random, independent, and identically distributed realization from the constructs' distribution (across both constructs and individuals). Under the Bayesian paradigm, the hypotheses linking the constructs are random variables, with distributions reflecting uncertainty. The associations are, therefore, probabilistic in nature. Each individual response is a conditionally independent realization of the underlying latent theme represented by the individual-specific construct, with those linked through also latent associations.

The proposed Bayesian approach allows for joint inference of constructs and latent associations, where the output is their joint posterior distribution (and the multivariate, correlated uncertainty about those). Evidence of associations is assessed based on whether the majority of the posterior mass for the corresponding latent association parameter is situated away from zero. Credible intervals are frequently formulated similarly to frequentist confidence intervals but have different interpretations [128]. A posterior 95% credible interval for a parameter of, for example, (0.383, 0.669) would indicate that a substantial amount of our knowledge about the parameter, given the information available in the data and any prior knowledge about the parameters, aligns with positive associations between the latent constructs.

### 3.5. Results

We summarize their results through traditional measures, including the posterior mean, 95% credible intervals, and a conclusion based on the strength of evidence about the latent associations. This is reported in Table 4 and in Figure 2. There is strong evidence of positive associations for all hypothesized relationships, with the majority of the posterior mass lying in positive territory in all cases. Full multivariate posterior distributions are obtained as part of the MCMC output, which shows good mixing as portrayed in Figure 3.

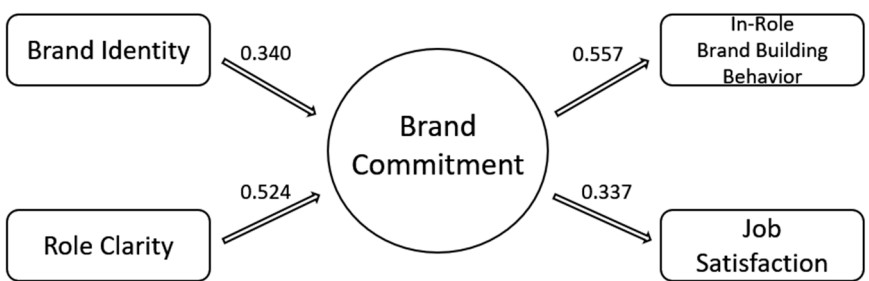

**Figure 2.** Research model results.

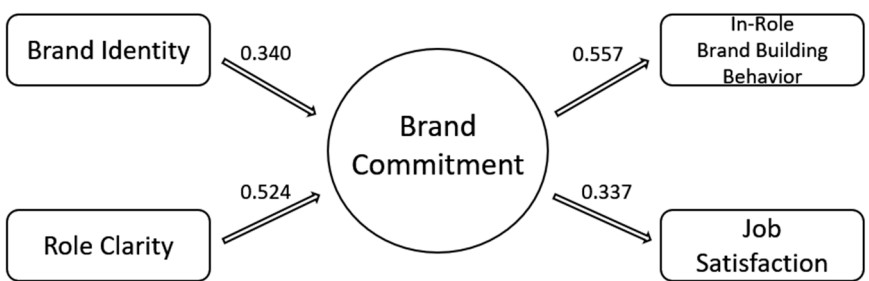

**Figure 3.** MCMC marginal posterior densities (**left**) and traceplots before thinning (**right**) for the parameters driving the hypothesized latent associations (beta1 = H1 in the top row through beta4 = H4 in the bottom row).

**Table 4.** Results from the MCMC regarding the four hypothesized latent associations.

|  | Hypotheses | Posterior Mean (Standard Deviation) | 95% Credible Interval | Evidence |
|---|---|---|---|---|
| H1 | Brand Identity → Brand Commitment | 0.340 (0.067) | 0.211–0.339 | Strong + |
| H2 | Role Clarity → Brand Commitment | 0.524 (0.072) | 0.383–0.669 | Strong + |
| H3 | Brand Commitment → IRBBB | 0.557 (0.068) | 0.425–0.691 | Strong + |
| H4 | Brand Commitment → Job Satisfaction | 0.337 (0.048) | 0.244–0.433 | Strong + |

## 4. Discussion

### 4.1. The Role of Brand Commitment for the Behavior of the Front-Lines Employees

Interestingly, role clarity had a much more substantial impact on brand commitment than brand identity. Brand identity clearly is a requirement for the development of brand commitment, but the results seem to indicate that FLEs place higher importance on understanding role expectations more clearly. This makes sense considering the existence of a close relationship between performance measurement and expectations in a job description that can impact employee compensation. The general assumption behind the proposed model proposed lies in the premise that other motivation methods might be available to organizations to mitigate the impact of this type of system. We proposed that the internalization of the brand's identity (H1) and a clear understanding of their role (H2) can cause FLEs to become more committed to the brand. These hypotheses were supported by substantial evidence from the results (posterior means of 0.340 and 0.524, respectively, with most of the posterior mass largely positive). These relationships should be leveraged by marketers working alongside HR to ensure the development of job descriptions fully aligned with branding expectations. These two findings are essential because they are variables that fall entirely under the control of organizations and can support the achievement of high levels of brand commitment.

Second, this study also examines how brand commitment influences FLE brand-oriented behavior and job satisfaction. Brand commitment is essential for organizations because a committed employee will be more likely to conform to behavioral expectations. Employee branding is examined from a normative perspective in the form of in-role behavior instead of the more frequently researched extra-role behavior. In-role behaviors are usually presented to employees through a job description document. These are presented as minimum requirements to employees by an organization against which employee performance is measured. In-role brand behavior existence is linked to a more transactional environment instead of extra-role brand behavior, which thrives in a more transformational environment [12]. A transactional environment is characterized by a system designed around metrics used to assess employee performance according to a set of key performance indicators typically found in an employee job description. This usually operates as a reward or punishment system, with employees being rewarded for doing the job as described or punished for the failure of doing so. In order to assess these relationships, we suggested that brand commitment had a positive impact on IRBBB (H3) and FLE job satisfaction (H4). Both hypotheses were supported (posterior means of 0.557 and 0.337, respectively).

### 4.2. The Relation between Brand Commitment and Open Innovation

Consumers understand brand innovation as the firms' ability to solve their problems in a new and valuable way. The extent that customers are likely to consider the interests of brands that focus on providing new and relevant solutions to customer needs under their own, the more likely customers are to engage with them in a positive way [129].

Relationship marketing has been sought as its ultimate goal: creating long-term relationships between client and company. However, service-dominant logic SDL has understood that this relationship can only be maintained as long as the company knows in depth the need and motivation of the client to co-create with them custom solutions [130]. In fact, using SDL and customer orientation has proved to be an essential tool to enhance

the volume and radicalness of firms' innovation, especially when involving FLE in the open innovation process [131].

The most direct and straightforward way for any company to contact its client is through the FLE. According to [132], the firm must become the principal actor who leads and "shake" the stakeholders. That is why this research has emphasized how to create brand commitment on the FLE, understanding that they are the canal to obtain information about the solutions sought by customers so that they can be transmitted as faithfully as possible to the company. This communication then becomes the central insight of the company to generate open innovation.

## 5. Conclusions

### 5.1. Theoretical Contribution

From an academic standpoint, this study provides additional empirical support for relationships discussed in the branding literature but have not been empirically tested in the retail sector in South America. It contributes to the marketing literature in many ways. It explores the impact of brand identity and role clarity on FLE brand commitment. Brand identity is described in the marketing literature as an internal organizational construct that can serve as a stable reference for both consumers and employees [133]. Associations linked to the brand start to form in people's minds once the brand identity is disseminated through communication mechanisms [134]. We argue that the same process occurs with FLEs, which explains the strong relationship observed between brand identity and brand commitment in this study, fully supporting H1. This relationship is meaningful because understanding a brand's identity is required to fully develop employee brand commitment.

In addition to the aforementioned contributions, this study presents a Bayesian framework for assessment of the relationships which has multiple advantages: (1) the hierarchical approach allows for exploration of latent relationships in a coherent, self-contained model where information borrowing is feasible across latent constructs; (2) the Bayesian paradigm is not reliant on large sample sizes, accounting for the natural uncertainty embedded in low samples; (3) although we used non-informative priors, if information was available about the parameters of interest, that information could be seamlessly incorporated in the analysis through informative priors; (4) it provides parameter-centered interpretability, where posterior distributions relate to the uncertainty of the parameters upon observing the data, as opposed to intermediate constructs/estimators; and (5) the outcomes of the Bayesian approach are full posterior distribution outcomes, which allow for better framing of the (joint) uncertainty around the hypotheses. This aligns with recent recommendations by the American Statistical Association to provide a more comprehensive depiction of results, rather than solely binary conclusions such as those traditionally built on *p*-values [126,127].

### 5.2. Practical Contribution

An employee branding program has the potential to contribute to the development of an environment where brand ambassadors can thrive and live the brand, thus positively impacting the customer experience. This is a challenging proposition that can be better supported through a better understanding of how FLE brand internalization occurs. This manuscript examines three factors that have relevance in this process and provides managers with a unique understanding of factors other than traditional tools and metrics employed by most companies to manage employee performance (i.e., job descriptions, scorecards, and key performance indicators (KPI)) in an emerging economy.

Although substantial recent literature focuses on employee brand-building process [94,135–137], its practical application still presents its own set of challenges. To start, brand identity requires a strong emotional appeal to connect with FLEs and generate internal brand commitment. For that to happen, the concept of brand identity must be managed through the organization's internal communications channels. Brand communication can help bridge the gap between the brand identity concept and employees' tasks. Therefore, we recommend that in order to address this gap, marketers must improve brand

knowledge by continually communicating the company-defined brand values and its vision of the brand image to FLEs. This can be accomplished by working together with the HR department and leveraging the organizational message systems controlled by it. HR departments can also further support the employee branding effort by incorporating brand elements in their processes, such as recruitment, compensation, training and development, and performance management systems. This step becomes critical to the development of job descriptions that incorporate branding guidelines determined by the marketing department that can improve employee role clarity. The marketing and HR departments must become partners in this process to avoid leaving programs that could support brand-orientated practices isolated under HR. Organizations can also improve employee role clarity and improve brand understanding by executing a hierarchical mapping of brand signals within the organization [138].

The prescription of in-role brand-building performance standards presents a significant challenge to managers [139]. The natural first step of this process would be developing brand identity guidelines for FLEs. These guidelines should be supported through internal training sessions alongside the human resources department in order to properly align brand requirements with other existing requirements in job descriptions [58]. Properly developed guidelines supported by FLE training can serve as a support pillar to the performance of prescribed brand behaviors that can impact how customers perceive the brand during a sales encounter. A real-world execution of these concepts was presented in a study of customers test-driving different Audi automobiles in the Netherlands and Denmark [58]. The authors selected Audi for this research project because of a strong belief of its senior leadership in the beneficial correlation between service experience and sales. A robust organizational belief existed in the organization that this correlation was an important step to help with their brand identity differentiation process. This belief led to the development of an internal concept named "The Audi Way" [58] (p. 168), aiming to develop a brand-building attitude shared by all members of the organization and explicitly focusing on FLEs. Ref. [58] reported results that showed that the investment in the FLE brand-building had a positive impact on both brand perception and customer satisfaction. We argue that the same premise can be applied to retailers.

In order to have satisfied customers, organizations must first have satisfied employees [140]. Job satisfaction is affected by job ambiguity and can be mediated by supervisor support, which should monitor the extrinsic and intrinsic sources of job satisfaction to which employees are subjected [119]. The main consequence of job dissatisfaction is employee turnover. Managers have widely acknowledged this issue since employee abandonment, and lack of organizational stability can significantly increase the costs associated with the orientation and training of new hires. Therefore, turnover can also impact organizational productivity. It is not surprising, then, that there has been a joint effort from organizational psychologists and researchers to identify clear antecedent factors that can lead to employee turnover. This knowledge can lead managers to institute more specific measures designed to prevent turnover [120]. Our findings point to the improvement of brand commitment as a possible way to improve job satisfaction. We argue that FLEs who can develop a strong association with the identity of their organization can also develop a more positive attitude towards their job. This can lead to increased recognition of the company's efforts directed at its employees. Consequently, FLEs may become more enthusiastic about providing additional effort than what is required beyond their current job description [141]. Additionally, employees that express satisfaction with their current jobs are more prone to fully accept and embody organizational brand values [142], leading them to engage in brand-oriented behaviors in exchange for organizational actions that improve their satisfaction with the job [143].

### 5.3. Limitations and Future Research

First, due to the cross-sectional nature of this study, it is essential to highlight that there is no statistical evidence of causality. Even though a theoretical rationale was pro-

vided to support the relationships examined and their direction, future research should attempt to replicate and extend this study by using longitudinal data to examine the causal relationships among focal constructs in the proposed model. Another significant limitation of this research was the use of a single retailer in one geographic location. This was mainly due to the difficulty involved in soliciting the participation of large retail organizations in research studies of this nature. There is always a level of reluctance to participate due to the investment required from them in terms of time away from front-line employees' sales floor. As is the case in such scenarios, further studies should be conducted in other retail formats and in different countries to generalize findings. Also, the data was collected through voluntary participation, hence potentially biasing the results if there are substantial differences between those volunteering and those who chose not to participate. Another significant limitation of this study is that it was restricted to a single country in South America, raising the question about the geographical generalizability of its findings. Therefore, being the first study of this kind in South America, it would be advantageous to conduct future research to establish whether the results presented here are inherent to the retail organization's format or if similar results could be generated in other retail formats. Another interesting avenue of future research is the comparison of these results with results from similar retailers in other South American markets. Finally, it would also be valuable to explore whether retail firms can distinguish in-role from extra-role behavior.

This research article explored the impact of brand identity and role clarity on FLE brand commitment in the retail sector and how brand commitment can support the engendering of brand-oriented behavior and lead to increased job satisfaction among front-line employees, as proposed in Figure 1. The findings presented here can provide managers with ideas to help develop an employee branding program that can produce more cable brand ambassadors. Furthermore, the proposed lines of research discussed above provide compelling ideas to explore further the relationships examined in this paper in different contexts.

**Author Contributions:** Data curation, E.t.H. and G.M.; formal analysis, E.t.H. and G.M.; writing—original draft, J.R.S.; writing—review and editing, N.P.-G. and M.V. All authors have read and agreed to the published version of the manuscript.

**Funding:** This research received no external funding.

**Institutional Review Board Statement:** The study was conducted according to the guidelines of the Declaration of Helsinki and approved by the Ethics Committee of CESA—Colegio de Estudios Superiores de Administración, project number 32004.

**Informed Consent Statement:** Informed consent was obtained from all subjects involved in the study.

**Data Availability Statement:** The data presented in this study are available on request from the corresponding author.

**Conflicts of Interest:** The authors declare no conflict of interest.

## Appendix A

Let $i$ denote each of the individuals, and j denote each of the Likert-based items across all constructs $k$ ($j = 1, \ldots, J(k)$). Index s denotes the five Likert categories ($s = 1, \ldots, 5$). Each item is represented as $Y$, with sub-indices for the construct, item, and respondent.

The latent array p denotes the latent Likert scale probabilities, with latent factors denoted as $Z$, and corresponding latent means $\mu$. Latent association parameters linking the constructs are denoted by the parameter vectors $\alpha$ (levels) and $\beta$ (linear associations representing the hypotheses in this study). The latent threshold matrix in the multinomial ordered representation is denoted as $\tau$. For a complete description of these types of models from a Bayesian standpoint, see [144].

The function $\varphi$ denotes the standard Gaussian cumulative density function, while $MN(p)$ denotes the multinomial probability mass function with probability vector $p$. $N(m,s)$

denotes the Gaussian probability density function with mean m and standard deviation s, and *Ga(a,b)* denotes the Gamma probability density function with parameters *a* and *b*.

Following the notation and approach in [122,123], the different hierarchies of the model have the following form:

$$Y_{i,j,k} \sim MN(p_{i,k})$$

$$p_{i,k,1} = \Phi(-Z_{i,k})$$

$$p_{i,k,s} = \Phi(\tau_{k,s} - Z_{i,k}) - \Phi(\tau_{k,s-1} - Z_{i,k}) \, s = 2, 3, 4$$

$$p_{i,k,5} = 1 - \sum_{s=1}^{4} p_{i,k,s}$$

$$Z_{i,k} = N(\mu_{i,k}, 1)$$

$$\mu_{i,k} = \alpha_k \, k = 1, 2$$

$$\mu_{i,3} = \alpha_3 + \beta_1 Z_{i,1} + \beta_2 Z_{i,2}$$

$$\mu_{i,4} = \alpha_4 + \beta_3 Z_{i,3}$$

$$\mu_{i,5} = \alpha_5 + \beta_4 Z_{i,3}$$

$$\beta_k, \alpha_k \sim N(0, 0.001)$$

$$\tau_{i,1} \equiv 0$$

$$\tau_{i,k} = \sum_{s=2}^{k} g_{i,s} \, k = 2, \dots, 4$$

$$g_{i,s} \sim Ga(0.001, 0, 0.001)$$

The model interpretation can be described through the model hierarchies:

1. The observed Likert responses are realizations of multinomial distributions with individual- and construct-specific probability vectors. This allows for both intra- and inter-respondent heterogeneity while accounting for intra-construct correlations;
2. The aforementioned probability vectors are mapped through a probit model into areas in the real line. While the true opinions are latent and unknown (Z), they are observed (Y) with some error;
3. The model hypotheses provide the associations between latent constructs rather than at the observed level. Associations will be observed with noise;
4. The final hierarchical level is composed by the prior densities, which are non-informative. If prior information is available, it can be incorporated into those prior densities. The results were not sensitive to the choice of hyperparameters for these priors.

The model was run for 100,000 iterations using OpenBUGS [145], with a burn-in of 10,000 iterations and a thinning of every 5 iterations. The results shown in Table 4 summarize the posterior distribution for the key parameters beta. Figure 3 provides further graphical details of the MCMC output for the key parameters of interest.

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
