# Peer review of "The Role of Brand Commitment in the Retail Sector: The Relation with Open Innovation"

_2199-8531, doi:10.3390/joitmc7020154_

Round 1

Reviewer 1 Report

General considerations

This article addresses a very pertinent topic regarding the integration of key stakeholders in the retail sector to improve the image of companies in the market through consumer satisfaction, a key factor for business success, even more in an unprecedented current context, in when economies quickly need to grow efficiently and effectively. The framework proposed with the study carried out in this investigation is an asset in contributing to the achievement of such purpose.

Grammar and spelling

The whole article it's well written in terms of grammar and spelling.

Article structure

The structure of the article is well elaborated, with no flaws detected in the numbering of the sections and subsections presented, as well as in the subtitle numeration of figures and tables.

Referencing

There are references that have a comma before the year of work (namely, for example, on lines 29, 688, 721, 727 and 755), and others that do not. This aspect should be standardized throughout the article.

Title, Abstract and Keywords

  • The title is appealing and contains key information about the article.
  • The abstract is well constructed and, refers to the focus of the article and highlights the main conclusions that were obtained in the investigation
  • The keywords are appropriate.

Section 1 (Introduction)

The introduction is very well elaborated, the ideas are very well explained and all articulated with each other. Also presented is the focus of the investigation and the most impactful conclusions obtained, as well as an indication of future work. However, a paragraph could be presented at the end, with a brief indication of the sections that make up the entire article, and not just the following one.

Section 2

  • All subsections of section 2 are very well explained and the ideas are exposed in a very clear and articulated way. But figure 1 seems to be extremely poor to illustrate the theoretical framework proposed by the authors;
  • There is also no link between hypothesis H1, H2, H3 and H4 that identify the key idea of each sub-chapter for integration in the proposed framework. Thus, it seems pertinent to propose a reassessment of the structure of figure 1, which does not live up to the exposed text, creating a big gap between the textual part and the schematic shown in figure 1.
  • The subtitle regarding figure 1 is completely generic, it seems pertinent to mention which conceptual model is being presented in a concrete way.
  • Figure 1 presents a schematic that lacks an explanation, which can be done in a short paragraph.

Sections 3 and 4

This sections are very well prepared: the collection and processing of data is very suitable for the study carried out, as well as the presentation and discussion of the results. However, the need for incrementing figure 1 is more emphasized, which now has quantitative information on the four hypotheses H1 to H4.

Section 5

This section establishes very well the interface between this investigation and the business environment, the text being very well explained. However, possibly a figure on this interface could be included to illustrate the text that is presented with high accuracy and quality.

Section 6

Section 6 is very well prepared, indicating well the limitations and future work inherent to this investigation. But, possibly it would be enriching to put a brief paragraph with a conclusion directed to the framework proposed by the authors.

Appendix

The appendix is very well structured and presents the necessary complements to the study developed in this investigation.

References

The list of references is well prepared, the number of references is appropriate to the depth of the theme's approach in the article and the text is very well referenced. The references are strong in the scope of the investigation.

Author Response

Thank for you review and recommendations. Please find our comments in the attached file. 

Reviewer 2 Report

Interesting - seems to be a solid methodology with useful results, clearly commicated, with implications for scholars and managers.

Actually, it was in the Managerial Implications section that I thought of why managers might care about this (more than some might think) -- and why researchers of corporate responsibility (sustainability) would cite and build on this research:

Namely, company branding and messaging is now part of how business leaders shake stakeholders our of complacency as part of their effort to co-create sustainable value - BOTH employees and (potential or existing) consumers (and others in society) in many contexts.

I would state this as one reason that (also) scholars of CSR (or sustainability) and managers would care about your findings! Here is the citation about business leaders shaking stakeholders: 

Sulkowski, A. J., Edwards, M., & Freeman, R. E. (2018). Shake Your Stakeholder: Firms Leading Engagement to Cocreate Sustainable Value. Organization and Environment.

Author Response

Thank you for your review and recommendations. Please find out comments in the attached document. 
